# Combining the Sensitivity of LAMP and Simplicity of Primer Extension via a DNA-Modified Nucleotide

**Moritz Welter and Andreas Marx \***

Department of Chemistry & Konstanz Research School Chemical Biology, University of Konstanz, 78457 Konstanz, Germany; moritz.welter@uni-konstanz.de

**\*** Correspondence:andreas.marx@uni-konstanz.de ; Tel.: +49-7531-88-5139

**Abstract:** LAMP is an approach for isothermal nucleic acids diagnostics with increasing importance but suffers from the need of tedious systems design and optimization for every new target. Here, we describe an approach for its simplification based on a single nucleoside-5′-O-triphosphate (dNTP) that is covalently modified with a DNA strand. We found that the DNA-modified dNTP is a substrate for DNA polymerases in versatile primer extension reactions despite its size and that the incorporated DNA indeed serves as a target for selective LAMP analysis.

**Keywords:** modified nucleotide; DNA polymerase; LAMP; primer extension

## 1. Introduction

Despite the widespread use of PCR-based amplification, the drawback of this technology, however, is its need for temperature cycling. Many attempts have been made to develop isothermal amplification methods that do not require heating of the double-stranded nucleic acid for the separation of templates [1]. These methods include strand-displacement amplification (SDA) [2], rolling circle amplification (RCA) [3], and helicase-dependent amplification (HDA) [4]. Another important method for nucleic acids diagnostics is loop mediated isothermal amplification (LAMP) [5,6]. LAMP relies on auto-cycling strand displacement DNA synthesis that is performed by a DNA polymerase with high strand displacement activity and a set of two specially designed inner and two outer primers (for details of the method, see Supplementary Materials Figure S1 in the ESI).

The reaction can be monitored by e.g., the addition of dyes used for nucleic acid staining [7] or turbidity analysis of precipitating magnesium phosphate [8]. Furthermore, the use of low-buffered reaction mixtures allows amplification monitoring with pH sensitive indicator dyes by the naked eye, as during the DNA polymerase reaction a proton is released for each nucleotide incorporation [9]. LAMP assays have also been adapted for many applications e.g., to cover genotyping and RNA detection [10–13]. However, in order to work as intended, the primers required for the amplification have to be carefully designed to meet particular requirements in regards to their melting temperature, spacing, and concentrations [5]. Thus, the design of suitable LAMP primers has to be optimized for every target that can be tedious, even when done with specific design software. While setting up the LAMP reaction, further problems are described such as the amplification of non-template controls, which drastically impedes the reliability of LAMP assays [14,15].

## 2. Materials and Methods

*General*

All reagents and solvents were obtained from Sigma-Aldrich (Darmstadt, Germany) and used without further purification. All synthetic reactions were performed under an inert atmosphere. Flash chromatography was performed using Merck silica gel G60 (230–400 mesh, Darmstadt,

Germany) and Merck precoated plates (silica gel 60 F254) were used for TLC. Anion-exchange chromatography was performed on an Äkta Purifier (GE Healthcare, Chicago, Il, USA) with a DEAE Sephadex™ A-25 (GE Healthcare Bio-Sciences, (GE Healthcare, Chicago, Il, USA) column using a linear gradient (0.1 M–1.0 M) of triethylammonium bicarbonate buffer (TEAB, pH 7.5). Reversed phase high pressure liquid chromatography (RP-HPLC, Shimadzu, Kyoto, Japan) for the purification of compounds was performed using a Shimadzu system having LC8a pumps and a Dynamax UV-1 detector (RP-HPLC, Shimadzu, Kyoto, Japan). A VP250/16 NUCLEODUR C18 HTec, 5 µm (Macherey-Nagel, D) column and a gradient of acetonitrile in 50 mM TEAA buffer were used. All compounds purified by RP-HPLC were obtained as their triethylammonium salts after repeated freeze-drying. NMR spectra were recorded on Bruker Avance III 400 ($^1$H: 400 MHz, $^{13}$C: 101 MHz, $^{31}$P: 162 MHz, Billerica, MA; USA) spectrometer. The solvent signals were used as references and the chemical shifts converted to the TMS scale and are given in ppm ($\delta$). HR-ESI-MS spectra were recorded on a Bruker Daltronics microTOF II. KlenTaq DNA polymerase was expressed and purified as described before. [16] T4 polynucleotide kinase (PNK) was purchased from New England BioLabs (Ipswich, MA; USA). [$\gamma$-$^{32}$P] ATP was purchased from Hartmann Analytics (Braunschweig, Deutschland) and natural dNTPs from Thermo Scientific (Waltham, MA, USA).

**Synthesis of nucleoside triphosphate 2.** To a solution of 18.9 µmol (10.3 mg) 5-(aminopentynyl)-2′-deoxyuridinetriphosphate tetrabutylammonium salt (**1**) [16] in 1 mL DMF, 94.5 µmol (5 eq., 9.5 mg) of NEt$_3$ were added. In parallel, 37.8 µmol (2 eq, 11.2 mg) of 16- azidohexadecanoic acid, 94.5 µmol (5 eq, 9.5 mg) NEt$_3$ and 37.8 µmol HATU (2 eq., 14.4 mg) were dissolved in 1 mL DMF and stirred for 30 min. Both mixtures were then combined and stirred at room temperature for additional 12 h. The solvent was removed under reduced pressure and the residual oil was subjected to C18-RP-HPLC (95% 50 mM triethylammonium acetate (TEAA) buffer to 100% MeCN). 2 was obtained in 58% yield as determined by Nanodrop ND1000 spectrometer with $\varepsilon$ (290 nm) = 13,300 M$^{-1}$·cm$^{-1}$. The compound was diluted in MilliQ water and kept as a 10 mM stock solution at −20 °C.

Analysis of **2**: $^1$H NMR (400 MHz, Methanol-d4) $\delta$ 8.01 (s, 1H, H-C(6)), 6.26 (t, $^3J$ = 6.8 Hz, 1H, H-C(1′)), 4.63–4.57 (m, 1H, H-C(3′)), 4.34–4.26 (m, 1H, H-C(5′a)), 4.23–4.17 (m, 1H, H-C(5′b)), 4.11–4.06 (m, 1H, H-C(4′)), 3.29 (t, $^3J$ = 6.9, 2H, H-C(L16)-), 3.22 (q, $^3J$ = 7.1, 17H, –CH2CH2CH2NH-, Et3N,), 2.46 (t, $^3J$ = 6.9 Hz, 2H, H-C(L2)), 2.32–2.25 (m, 2H, H-C(2′)), 2.1 (t, $^3J$ = 7.5, 2H, –CH2CH2CH2NH-), 2.20 (t, $^3J$ = 7.6 Hz, 2H, H-C(L2)), 1.79 (p, $^3J$ = 6.8 Hz, 2H, –CH2CH2CH2NH-), 1.60 (p, $^3J$ = 6.8 Hz, 4H, H-C(L3+15)), 1.45–1.27 (m, 51H, H-C(4-14), Et3N). 31P NMR (243 MHz, Methanol-d4): $\delta$ = -10.44 (d, $^2J$ = 20.5 Hz), −11.33 (d, $^2J$ = 21.3 Hz), −23.72 (t, $^2J$ = 21.3 Hz). HR-ESI-MS (*m/z*): [M − H]$^-$ = calcd: 827.2552; found: 827.2562.

**Preparation of dT$^{15LAMP}$TP.** The split LAMP target sequence was ligated using T4 DNA ligase and a splint oligonucleotide. 1 nmol of LAMP_TARGET_A and LAMP_TARGET_B (10 µM, Biomers.net) were mixed with 2 nmol (20 µM) of the splint oligonucleotide in a total volume of 98 µL of 1x T4 ligase buffer provided by the manufacturer (NEB). The mixture was heated to 95 °C for 2 min and slowly cooled down to 25 °C. Subsequently, 2 µL of T4 Ligase (800 U) were added and the reaction was incubated at 16 °C overnight. The mixture was then diluted to 200 µL with MilliQ water and subjected to 95 °C for 5 min. Ion-exchange HPLC was performed at 85 °C column temperature using 100 µL of the solution on an analytical HPLC system with a semi-preparative Thermo Scientific™ Dionex™ DNAPac™ PA100 column and a gradient from IEX-HPLC buffer A (25 mM Tris-HCl, pH 8) to IEX-HPLC buffer B (25 mM Tris-HCl, 0.5 M sodium perchlorate, pH 8). Peaks demonstrating an absorbance at $\lambda$ = 260 nm were collected and pooled in Amicon 4 centrifugal filters. After repeated washing with MilliQ water, the ligated LAMP target was transferred to a 1.5 mL reaction tube and absorbance was measured by NanoDrop ND-1000 spectrometry at 260 nm with $\varepsilon$ (403 nm) = 1,752,400 M$^{-1}$ cm$^{-1}$. To conjugate the 5′-DBCO labeled LAMP target with compound **2**, the above generated oligonucleotide was incubated with 10 eq of the nucleotide in 1x PBS (pH 7.4) overnight. IEX-HPLC and Amicon purification were repeated to yield **dT$^{15LAMP}$TP**.

**Primer extension (PEx) in solution with dT$^{15LAMP}$TP.** To 1x polymerase buffer (50 mM Tris-HCl, 16 mM ammonium sulfate, 2.5 mM magnesium chloride, 0.1% Tween 20, pH 9.2), 150 nM 5′-$^{32}$P-labeled primer and 200 nM template were added. The mixture was annealed at 95°C for 5 min. Subsequently, DNA polymerase was added (100 nM KlenTaq DNA polymerase) and the reaction was started by addition of the dNTP (1 µM final concentration). Time points were collected by quenching 2 µl of the reaction mixture with 10 µl stopping solution (80% *v/v* formamide, 20 mM EDTA, 0.025% *w/v* bromophenol blue, 0.025%

*w/v* xylene cyanol). Denaturing polyacrylamide gels (9%) were prepared by polymerization of a solution of urea (8.3 M) and bisacrylamide/acrylamide (9%) in TBE buffer using ammonium peroxodisulfate (APS, 0.08%) and *N,N,N′,N′*-tetramethylethylene-diamine (TEMED, 0.04%). Immediately after addition of APS and TEMED, the solution was filled in a sequencing gel chamber (Bio-Rad) and left for polymerization for at least 45 min. After addition of TBE buffer (1x) to the electrophoresis unit, gels were pre-warmed by electrophoresis at 100 W for 30 min and samples were added and separated during electrophoresis (100 W) for approximately 1.5 h. The gel was transferred to Whatman filter paper, dried at 80 °C in vacuo using a gel dryer, and exposed to an imager screen. Readout was performed with a molecular imager FX.

**LAMP assay in solution.** To avoid contaminations, all LAMP reactions were pipetted with Biosphere filter tips. The LAMP target used for the positive controls was pipetted with a second pipette set. Initial LAMP reactions were performed according to the conditions reported by Tanner and co-workers [9] with 8 U Bst 2.0 WarmStart® DNA Polymerase (NEB) DNA polymerase, 1x SYBR I, 0.2/0.4/1.6 μM LAMP primers (outer/loop/inner), 10 nM 5′-DBCO LAMP target (positive control), 350 μM/dNTP, 65 °C in 1x isothermal amplification buffer (NEB) with 8 mM MgSO4. LAMP reactions in optimized conditions were carried out using 200 μM dNTPs, 0.2/0.4/1.6 μM primers (outer/loop/inner), 1x SYBR I, 4 U of Isotherm2G DNA polymerase (myPOLS Biotec, Konstanz, Germany) and 0.1 nM 5′-DBCO LAMP target (positive control) in a total of 10 μL of 1x Isotherm2G buffer at 55 °C for the indicated amount of time. Fluorescence of SYBR I was measured in 1 min intervals minute in a Bio-Rad CFX384 Touch™ Real-Time PCR Detection System. Following the amplification, melting point measurement was carried out with a gradient from 55 °C to 95 °C in 0.5 °C steps.

**Primer extension (PEx) and LAMP assay using immobilized primers.** Pierce Streptavidin coated 8-well strips (ThermoFisher, Waltham, MA, USA) were washed twice with 200 μL of 1x plate washing buffer. Subsequently, 1 μL of 500 μM 5′-biotin BRAF primer in 100 μL 1x PBS buffer were added to each well. After 15 min of incubation at room temperature, the primer solution was removed and 200 μL of 1 mM (D)-+-biotin in 1x PBS were added. After 5 min of incubation, the liquid was removed and the wells were washed once with 200 μL of PBS buffer and twice with 200 μL of 1x KTq reaction buffer. Following this, 50 μL of PEx reaction mixture (100 nM KlenTaq DNA polymerase, 200 nM BRAF template, 200 nM **dT$^{15LAMP}$TP** in a total of 50 μL 1x reaction buffer (50 mM Tris-HCl, 16 mM ammonium sulfate, 2.5 mM magnesium chloride, 0.1% Tween 20, pH 9.2) were applied, the wells were sealed with PCR foil seal and incubated at 55 °C (measured with a digital thermometer) in a shallow water bath on a thermal block for 30 min. The supernatant was removed and the wells were washed first twice with 100 μL 1x reaction buffer, then three times with 1x PBS buffer and finally rinsed for 2 min under a water tap with MilliQ water. All liquid was removed and the wells were finally washed with 200 μL of 1x Isotherm2G DNA polymerase buffer. 50 μL of LAMP reaction mixture (200 μM dNTPs, 0.2/0.4/1.6 μM primers (outer/loop/inner), 1x SYBR I, 4 U Isotherm2G DNA polymerase) were employed in each well. For the positive control, 2 nM 5′-DBCO LAMP targets were added. The wells were incubated at 55 °C for the primer extension. Amplification was stopped by rapidly cooling the wells to 0 °C in an ice bath. Samples were instantly collected and run on a 2.5% agarose gel. The gel was read out using GelRed staining under UV light on a Chemidoc™ XRS system (Bio-Rad, Hercules, CA, USA).

## 3. Results

Here, we describe an approach towards the simplification of LAMP reactions based on a nucleoside-5′-*O*-triphosphate (dNTP) that is modified with a DNA strand serving as a LAMP target (Figure 1).

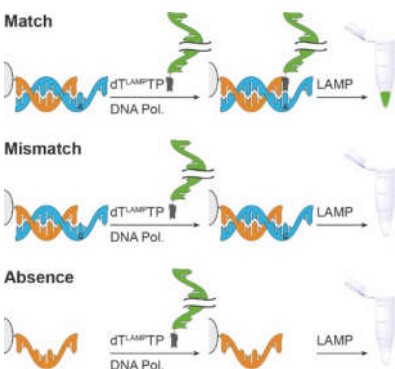

**Figure 1.** Depiction of the approach explored in this study. A primer is immobilized on a solid support via biotin-streptavidin interaction. After annealing of the template sequence, DNA polymerase and the LAMP template-modified dNTP (dT$^{LAMP}$TP) are added. After incubation, the unbound conjugate is removed by repeated washing and the LAMP reaction mixture is added afterwards. Only in cases where matched primer template duplexes are present (top), the LAMP reaction is expected to be positive.

The LAMP template-modified dNTP (dT$^{LAMP}$TP) may be a substrate for DNA polymerases in sequence selective primer extension reactions on solid support. Single nucleotide incorporation is highly sequence selective [17,18] and allows discrimination of single nucleotide variations by the mere difference of incorporation efficiencies of a matched versus mismatched nucleotide. In turn, the immobilized LAMP template (green, Figure 1) can be targeted by strand displacement-proficient DNA polymerases in LAMP reactions. This approach holds promise offering the advantage that—in principle—with one single LAMP sequence an infinite number of targets can be analyzed without tedious redesign of the LAMP sequence, since the immobilized primer strand (orange, Figure 1) is responsible for the sequence selective capture of the target (blue, Figure 1).

The approach depicted above is based on the covalent connection of the LAMP template to a nucleotide. A LAMP template, however, has to harbour six distinct binding sites in fixed spacing. Typical sequences hence consist of around 200 or more nucleotides (nt), which vastly exceed the oligonucleotide-modified nucleotides that have been reported for successful incorporation and which were modified with up to 40 nucleotides [19]. We chose a LAMP template (sequence see Table 1) with a 245 nt sequence taken from the genome of the Lambda phage [9,20]. The sequence was shortened by 61 nucleotides in the middle area and split up into two halves with lengths of 91 nt and 93 nt. For conjugation to the nucleotide, the oligonucleotide representing the 5′-end of the target sequence was equipped with a 5′-dibenzocyclooctyne (DBCO) modification (Figure 2). The 3′-half of the sequence was phosphorylated on its 5′-end to allow splint ligation with T4 DNA ligase, which was carried out at 16 °C in presence of two equivalents of a 30nt splint.

**Table 1.** Employed DNA sequences.

| | |
|---|---|
| LAMP Backward Inner Primer | 5′-d (GAG AGA ATT TGT ACC ACC TCC CAC CGG GCA CAT AGC AGT CCT AGG GAC AGT) |
| LAMP backward loop primer | 5′-d (ACC ATC TAT GAC TGT ACG CC) |
| LAMP backward outer primer | 5′-d (GGA CGT TTG TAA TGT CCG CTC C) |
| LAMP forward inner primer | 5′-d (CAG CCA GCC GCA GCA CGT TCG CTC ATA GGA GAT ATG GTA GAG CCG C) |
| LAMP forward loop primer | 5′-d (CTG CAT ACG ACG TGT CT) |
| LAMP forward outer primer | 5′-d (GGC TTG GCT CTG CTA ACA CGT T) |
| LAMP_Splint | 5′-d (GGC TGG CTG TCC AGT GAG AGA ATT TGT ACC) |
| LAMP_Target_A | 5′-DBCO-d (GGA CGT TTG TAA TGT CCG CTC CGG CAC ATA GCA GTC CTA GGG ACA GTG GCG TAC AGT CAT AGAT GGT CGG TGG GAG GTG GTA CAA ATT CTC TC) |

| LAMP_Target_B | 5′-P-d (ACT GGA CAG CCA GCC GCA GCA CGT TCC TGC ATA CGA CGT GTC TGC GGC TCT ACC ATA TCT CCT ATG AGC AAC GTG TTA GCA GAG CCA AGC C) |
|---|---|
| 5′-biotin BRAF primer | 5′-biotin-d (TTT TTT TTT TTT TTT TTT TGA CCC ACT CCA TCG AGA TTT C) |
| BRAF template (DNA) | 5′-d (TGC CTG GTG TTT GGG AGA AAT CTC GAT GGA GTG GGT C) |

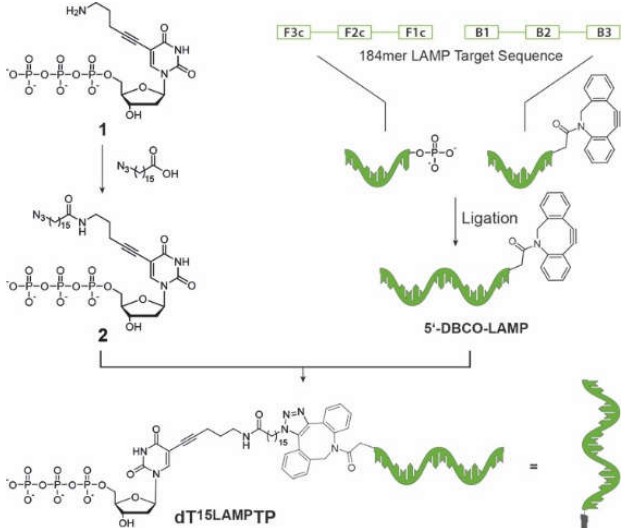

**Figure 2.** Synthesis of LAMP template-conjugated dT$^{15LAMP}$TP. Left: reaction conditions: 1, DMF, HATU, Et3N, rt, 12 h; right: the two modified halves of the LAMP template are ligated by splint ligation with T4 DNA ligase yielding a 5′-DBCO-modified 184mer. The click reaction between the azide-functionalized nucleotide and the LAMP template is carried out in PBS buffer at room temperature 12 h and the product is purified by ion-exchange HPLC.

To react with the 5′-DBCO modified oligonucleotide that harbors the LAMP sequence, an azide-functionalized nucleotide was prepared starting from the known dTTP analog 1 (Figure 2) [16]. We chose a linker length that has been demonstrated before to be suitable for appending large "cargo" to dNTPs without greatly compromising DNA polymerase activity [19]. Employment of 16-azidohexadecanoic acid [21], HATU, and Et3N in DMF yields compound 2. Conjugation of 2 and the LAMP template by strain-promoted 1,3-dipol cycloaddition (SPAAC) [22] was achieved in PBS buffer and the product dT$^{15LAMP}$TP was purified by ion exchange HPLC and centrifugal filtration.

Next, primer extension experiments were conducted with dT$^{15LAMP}$TP in comparison with natural dTTP and the dTTP derivative 2 with KlenTaq DNA polymerase using a template containing the B type raf kinase (BRAF) T1796A point mutation, which is strongly associated with carcinogenesis [23]. After incubation, samples of the primer extension (PEx) reaction were quenched and analyzed by denaturing polyacrylamide gel electrophoresis (PAGE). For dTTP, the expected shift for single nucleotide incorporation (Figure 3) was observed.

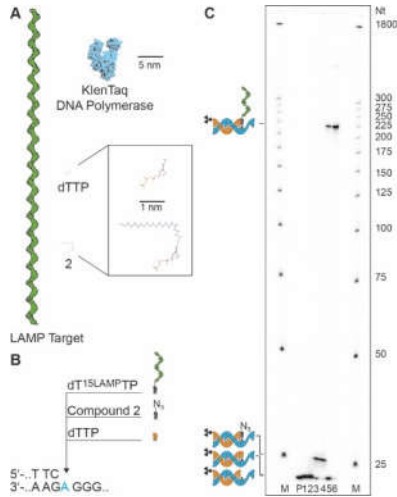

**Figure 3.** Primer extension experiment employing the dT[15LAMP]TP. (**A**) The LAMP target sequence, KlenTaq DNA polymerase, natural dTTP and compound 2 drawn to scale. (**B**) Partial sequence of the incorporation site in the BRAF sequence context and the three different nucleotides used in this experiment. (**C**) PEx experiment with KlenTaq DNA polymerase at 55 °C with 1 μM dT[15LAMP]TP. P: Primer, 1: dTTP, 1 min, 2: dTTP, 30 min; 3: Compound **2**, 1 min, 4: Compound 2, 30 min; 5: dT[15LAMP]TP, 1 min, 6: dT[15LAMP]TP, 30 min; M: Marker.

Processing of compound **2** and incorporation of the respective modified nucleotide into the nascent DNA strand led to a pronounced shift of the product by PAGE analysis due to the long alkyl chain-modification impeding migration through the gel matrix. Finally, the usage of dT[15LAMP]TP led to a very pronounced shift of the product in PAGE analysis, similar to that observed when protein-conjugated nucleotides and shorter oligonucleotide-conjugated nucleotides were used [19,21,24–26]. The band corresponding to the LAMP sequence-conjugated nucleotide runs at approx. 225 nt, which is consistent with the combined size of the 21-mer primer, the 184 nt LAMP sequence, and the connecting alkyl linker. Therefore, not only was the conjugation between the LAMP sequence and the nucleotide confirmed, but it was also shown that this DNA polymerase is able to incorporate nucleotides equipped with ssDNA, being considerably longer than the sequence context used for incorporation.

With the LAMP target-modified nucleotide dT[15LAMP]TP in hand, the LAMP reaction itself was optimized. The assay was conducted as reported in the original publication with Bst 2.0 DNA polymerase, 350 μM for each dNTP, and 0.2/0.4/1.6 μM primers (outer/loop/inner, respectively) at 65 °C and monitored by real-time SYBR green I fluorescence detection [9]. Using these conditions, the positive control containing 10 nM of the ligated LAMP target was amplified at a cycle quantification value (Cq) of 15.4, but all non-template controls (NTC) showed a similar amplification ranging from Cq 26.4 to 46.6 (Figure 4A). To ensure that this behavior was not caused by a contamination with LAMP target, the experiment was repeated several times with freshly prepared reagents. However, amplification within non-template controls was persistent in all runs. Similar issues were reported in other studies with false positive amplification in LAMP assays [14,15].

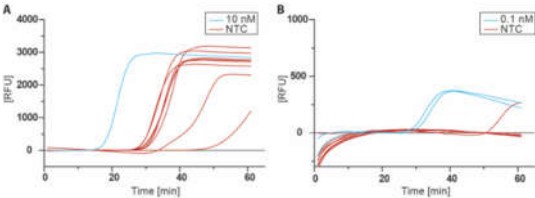

**Figure 4.** Real-time monitoring of the LAMP with SYBR green I using dT[15LAMP]TP. (**A**) LAMP Assay using the conditions reported by Tanner et al. [9] with Bst 2.0 DNA polymerase at 65 °C, 0.2/0.4/1.6 μM outer/loop/inner primers, 8 mM MgSO4 and 350 μM dNTP each. (**B**) Optimized conditions with

Isotherm2G DNA Polymerase at 55 °C, 0.2/0.4/1.6 µM outer/loop/inner primers, 2 mM MgSO4 and 200 µM dNTP each.

To overcome these issues, a screening for appropriate LAMP conditions was carried out including different DNA polymerases, incubation temperatures, primer ratios and concentrations of dNTPs, $Mg^{2+}$, SYBR green I and betaine. In the end, the assay conditions were changed to Isotherm2G DNA polymerase, 55 °C reaction temperature, 200 µM dNTP each with SYBR green I and remaining primer concentrations at their original levels (0.2/0.4/1.6 µM outer/loop/inner primer). Primers were denatured prior to the addition to the master mix in order to minimize the effect of primer dimers or the presence of any self-primed secondary structure. With the optimized conditions, 0.1 nM of the ligated LAMP target were detected at Cq 28 with no to minimal false positive reactions, which was amplified at sufficiently delayed time points (Cq 51, Figure 4B).

Having optimized the LAMP conditions, we next investigated immobilized primers in a primer extension of the LAMP target-modified nucleotide and subsequent LAMP reaction. Therefore, we used streptavidin coated 8-well plates as the solid phase to immobilize biotin-modified primer strands. The wells were first incubated with 5′ biotinylated primer in PBS and subsequently blocked with biotin. After several washing steps, 50 µL of primer extension reaction mixture containing 100 nM KlenTaq DNA polymerase, 1 µM dT$^{15LAMP}$TP, and 200 nM BRAF template was added (Figure 5, lane 1).

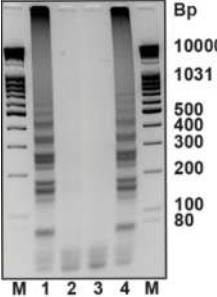

**Figure 5.** LAMP detection of DNA targets**.** A PEx reaction employing dT$^{15LAMP}$TP and KlenTaq DNA polymerase was performed on an immobilized primer in the presence or absence of the matched PEx template. Subsequently, the wells were washed and a LAMP reaction mixture was added. Samples of each well were taken and analysed on a 2.5% agarose gel. Picture colours were inverted to improve contrast. 1: 200 nM (10 pmol) matched BRAF template, 2: no BRAF template, 3: 200 nM BRAF template but natural dTTP instead of dT$^{15LAMP}$TP, 4: no PEx reactions but LAMP reaction spiked with LAMP target sequence as a positive control, M: marker.

Controls were set up without a BRAF template (lane 2) or with dTTP instead of the LAMP target modified nucleotide dT$^{15LAMP}$TP (lane 3). Following 30 min of incubation at 55 °C, the reaction mixture was removed. After the plates were washed intensively with 1x KlenTaq reaction buffer, PBS buffer, MilliQ water and 1× Isotherm2G reaction buffer, the LAMP reaction mixture was applied. Furthermore, an additional sample (Figure 5, lane 4) was treated equally as the other samples, but incubated in KlenTaq reaction buffer instead of a primer extension reaction mixture. Here, the LAMP reaction was spiked with 2 nM of LAMP target to serve as a positive control. The LAMP reaction was incubated at 55 °C for 20 min and stopped by rapid cooling of the plates in ice water. Samples were taken and directly subjected to analysis by agarose gel electrophoresis. Upon agarose gel electrography analysis, a ladder-like pattern of bands was observed for the positive reaction (Figure 5, lane 1). The pattern is consistent with the positive control in well 4, which proves the specific amplification starting from the LAMP target. No amplification was observed for both negative controls analysed in lanes 2 and 3. Hence, a LAMP reaction can be utilized to detect the presence of a PEx template using the LAMP target-conjugated dTTP derivative dT$^{15LAMP}$TP.

## 4. Conclusions

In summary, a shortened LAMP target sequence derived from the genome of the Lambda phage was generated and conjugated to an azide-functionalized dTTP derivative via click chemistry to yield dT[15LAMP]TP. Primer extension experiments with the LAMP conjugate revealed that KlenTaq DNA polymerase is able to incorporate the modified nucleotide into a primer strand in spite of the length of the attached oligonucleotide. Next, dT[15LAMP]TP was employed in primer extension reactions in solid phase in which the modification, if covalently connected to the primer, served as a reporter for the presence of the template in the primer extension reaction. Indeed, we found that amplification starting from the immobilized LAMP template was only observed if the template for the preceding primer extension was present in the reaction due to processing of dT[15LAMP]TP. Thus, the results demonstrate proof-of-concept that the robustness and simple setup of primer extension-based assays with the rapid and sensitive amplification of LAMP is feasible. The herein depicted results might advance and simplify LAMP-based applications.

**Supplementary Materials:** The following are available online at www.mdpi.com//2/2/29/s1, Figure S1: Nucleic acid amplification by LAMP. The two inner primers (green in Step I and orange in Step II) comprise of a site complementary to a sequence in the target oligonucleotide and a 5′ overhang that is complementary to a site within the elongated primer (F1c, B1c). After the inner primer is elongated, the outer primer (black) binds upstream of the inner primer at the target sequence and its elongation by the strand displacement DNA polymerase releases the prolonged inner primer (Step I). The procedure is then repeated at the other side of the released, elongated inner primer (Step II), generating a new sequence that is similar to the target sequence, but instead of the outer primer binding site, it is now on both sides equipped with a sequence complementary to an area inside the oligonucleotide (orange, Step III). Annealing of these complementary sequences will lead to a dumb bell-like structure in which first, the self-primed 3′ end is elongated by the DNA polymerase to open the dumb bell-end on the other side and second, the annealing and elongation of new inner primer releases the stem-loop generated in the first step (Step IV). Thus, a new self- primed 3′-end is formed with which the cycle of self-primed elongation and release by an inner primer is continued. In the end, a mixture of stem-loop like DNA concatemers and cauliflower-like structures with various repeat counts are obtained (Step V).

**Author Contributions:** Conceptualization, M.W. and A.M.; methodology, M.W. and A.M.; validation, M.W. and A.M.; formal analysis, M.W.; investigation, M.W.; resources, A.M.; writing—original draft preparation, M.W. and A.M.; writing—review and editing, M.W. and A.M.; visualization, M.W. and A.M.; supervision, A.M.; project administration, A.M.; funding acquisition, A.M. All authors have read and agreed to the published version of the manuscript.

**Funding:** This research was funded by DFG Deutsche Forschungsgemeinschaft, grant number MA 2288/16-2.

**Acknowledgments:** We thank Dr. Samra Ludmann for assisting in the preparation of the manuscript.

**Conflicts of Interest:** The authors declare no conflict of interest.

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
