# Peer review of "Combining the Sensitivity of LAMP and Simplicity of Primer Extension via a DNA-Modified Nucleotide"

_chemistry, doi:10.3390/chemistry2020029_

Round 1
Reviewer 1 Report
The manuscript by Welter and Marx describes a nice and potentially useful improvement of the known LAMP that alleviates the need for thermal cycling in nucleic acids amplificative detection techniques. A chemically modified LAMP target (by strain-induced azide-alkyne cycloaddition) is covalently linked to a solid support-bound primer, which is then submitted to LAMP conditions. The scientific approach is sound, and the relevant control experiments have been carried out, after condition optimization. The experimental part (as SI) is sufficiently clear to be reproduced, and the synthetic compounds have been appropriately characterized. Thus, I recommend publication of this manuscript. A possible improvement could, hoiwever, be brought to the ambiguius formulation of the paragraph beginning line 95: it is not completely clear what "emplyment of compound 2" means in terms of PAGE analysis (is it 2 itself, or a conjugate?).
Author Response
We have changed the corresponding paragraph to:
“Processing of compound 2 and incorporation of the respective modified nucleotide into the nascent DNA strand led to a pronounced shift of the product by PAGE analysis due to the long alkyl chain-modification of impeding migration through the gel matrix.”
Reviewer 2 Report
Welter and Marx have described findings that the primer extension-based assays employing loop-mediated isothermal amplification (LAMP) is feasible with simplification based on a single nucleoside-5’-O-triphosphate modification. The approach depicted in the manuscript is based on the covalent connection of the LAMP template to nucleotide. It, in general, follows described by Professor Marx laboratory as well as others protocol which involves strain-promoted azide-alkyne cycloaddition (SPAAC) reaction between 5’-DBCO modified oligonucleotide (cyloalkyne) that harbors the LAMP sequence and 5-azido-functionalized pyrimidine nucleotide (dTTP). The method advances and simplifies LAMP-based applications that justified publication. The manuscript is concise, well-written and literature quoted properly. Discussion and experimental part are written with enough details and experiments are well described and characterized. I recommended publication with very minor revisions.
- Can the authors comment on the length of linkage needed for 5-azido-functionalized dTTP? Has 16-azidohexadecanoic acid given optimal results in covalent connection to cyloalkyne attached to the LAMP sequence which bears distinct binding sites in fixed spacing? Did authors attempt covalent connection with azides bearing shorter or longer carbon chain?
- Statement regarding availability of Supplementary Materials is missing.
Author Response
“Can the authors comment on the length of linkage needed for 5-azido-functionalized dTTP? Has 16-azidohexadecanoic acid given optimal results in covalent connection to cyloalkyne attached to the LAMP sequence which bears distinct binding sites in fixed spacing? Did authors attempt covalent connection with azides bearing shorter or longer carbon chain?”
Response: We chose a linker length that have been demonstrated before to be suitable for appending large “cargo” to dNTPs without compromising DNA polymerase activity greatly [19].
“Statement regarding availability of Supplementary Materials is missing.”
Response: The statement has now been added.